# Increasing trends in regional heatwaves

S. E. Perkins-Kirkpatrick [1✉] & S. C. Lewis[2]

Heatwaves have increased in intensity, frequency and duration, with these trends projected to worsen under enhanced global warming. Understanding regional heatwave trends has critical implications for the biophysical and human systems they impact. Until now a comprehensive assessment of regional observed changes was hindered by the range of metrics employed, underpinning datasets, and time periods examined. Here, using the Berkeley Earth temperature dataset and key heatwave metrics, we systematically examine regional and global observed heatwave trends. In almost all regions, heatwave frequency demonstrates the most rapid and significant change. A measure of cumulative heat shows significant increases almost everywhere since the 1950s, mainly driven by heatwave days. Trends in heatwave frequency, duration and cumulative heat have accelerated since the 1950s, and due to the high influence of variability we recommend regional trends are assessed over multiple decades. Our results provide comparable regional observed heatwave trends, on spatial and temporal scales necessary for understanding impacts.

[1] Climate Change Research Centre, UNSW Sydney, Sydney, NSW, Australia. [2] School of Science, UNSW Canberra, Canberra, ACT, Australia.
✉email: Sarah.Kirkpatrick@unsw.edu.au

Defined as prolonged periods of excessive heat[1], heatwaves are a specific type of extreme temperature event. There are many adverse impacts of heatwaves, including on human health[2], agriculture[3,4], workplace productivity[5], wildfire frequency and intensity[6], and public infrastructure[7,8]. The inequality of heatwave impacts has been assessed[5,9], adversely affecting developing nations due to a lack of adaptive capacity, as well as varying cultural constraints. These impacts will increase under enhanced global warming, where more rapid heatwave trends will likely produce more severe and possibly irreversible impacts in some sectors[9–12].

There are multiple characteristics to heatwaves, including their intensity, frequency, duration, timing and spatial extent. Additionally, there are numerous ways to define each characteristic, a likely consequence of their expansive impacts as well as potential limitations of the datasets from which they are assessed. There is a general consensus that the intensity, frequency and duration of heatwaves have increased in the observational record, both regionally and globally[1,13–17]. However, all relevant studies have employed diverse metrics, analysed a subset of heatwave characteristics over selected regions, and/or used different extreme thresholds over varying time periods. This undermines a comprehensive understanding of how heatwaves have changed at regional and global scales. An all-inclusive and consistent assessment is imperative for establishing confidence in the general consensus suggested by previous disparate studies, as well as being an integral component in accurately assessing the occurrence of and changes in heatwave impacts.

This study provides such an assessment. While data availability limits our analysis to begin in 1950, we find that trends in the spatially consistent Berkeley Earth[18] observational dataset match well with common regions between the previously used quasi-global HadGHCND[19] dataset and Berkeley Earth is thus useful in furthering our understanding of global and regional changes in heatwaves. Heatwave frequency demonstrates the most significant increase across almost all regions, with nowhere experiencing a significant decrease. While average heatwave intensity displays little change, cumulative heatwave intensity increases at a similar rate to heatwave frequency. Moreover, the most intense heatwave seasons as defined by cumulative intensity generally occur post 2000. By assessing regional trends that commence from all years between 1950 and 2000, we find that the increasing trends of historical heatwave frequency, duration and cumulative intensity have accelerated over many regions. However, climate variability can be a large influence on heatwave trends over timescales of multiple decades, even in recent decades, which are more heavily influenced by enhanced anthropogenic climate change.

## Results

**Global trends in heatwave characteristics.** A previous attempt to address global observed heatwave changes[13] employed the HadGHCND[19] dataset and computed trends for 1950−2011. Although a plethora of metrics exist, we define heatwaves using a consistent methodological framework[1]. This is when at least 3 consecutive days are above a percentile-based extreme threshold over a 5-month warm season (see "Methods"). We assess changes in heatwave frequency, duration and intensity, and also trends in seasonal cumulative heat, briefly expressed as the sum of the temperature anomaly relative to the respective heatwave threshold, across all heatwave days within the season (see "Methods" for more detailed explanations of heatwave characteristics). The cumulative heat metric provides additional and relevant information for impacts[12] that is presented here at global and regional scales, alongside the more traditional heatwave metrics used in climate-based studies.

We primarily focus on the Berkeley Earth observational dataset[18] (see "Methods") and analyse trends both globally and over the Special Report on Extremes (SREX) regions[11] (see Supplementary Table 1). To explore the validity of Berkeley Earth in measuring heatwave trends, we compare the two datasets (Fig. 1) for heatwave frequency (Fig. 1a, b), duration (Fig. 1c, d), intensity (Fig. 1e, f) and cumulative heat (Fig. 1g, h; see "Methods") where trends span 1950−2014. Note that HadGHCND can only produce a quasi-global assessment of heatwave changes, due to large amounts of missing data over Central and South America, the Middle East, India, Indonesia, northern Canada and Greenland, inhibiting a true assessment of observed heatwave changes. In addition, the coarse resolution of 3.75° longitude by 2.5° latitude grid hampers an assessment of heatwave changes at the scales useful for understanding impacts. On the other hand, over the same time period Berkeley Earth provides much greater spatial coverage at a finer resolution (1° longitude by 1° latitude). Although the methods for constructing Berkeley Earth differ from HadGHCND, there is high agreement of the magnitude and significance of heatwave trends for common regions (Fig. 1). Agreement also occurs for trends commencing in subsequent decades (see Supplementary Figs. 1 −4). Despite which year heatwave trends commence, the level of agreement between the datasets remains high. Moreover, while trend magnitudes generally increase through time the significance decreases, due to the increasing influence of climate variability on heatwaves over shorter timescales[20,21], which is reflected by both datasets.

According to Berkeley Earth, heatwave frequency demonstrates the most widespread and significant increase of the characteristics analysed (Fig. 1b). Heatwave duration (Fig. 1d), although increasing, has significant trends restricted to South America, Africa, the Middle East, and Southwest Asia. Significant heatwave intensity trends (Fig. 1f) are non-existent for most of the globe, the exception being southern Australia and small areas of Africa and South America. On the other hand, significant cumulative heat trends (Fig. 1h) are comparable in space to heatwave frequency (Fig. 1b), with mainly positive magnitudes. The largest trends are seen over the Middle East and parts of Africa and South America, where the extra heat produced by heatwaves is increasing by 10 °C decade$^{-1}$. For most other areas with significant trends, cumulative heat increases by 2−6 °C decade$^{-1}$.

**Regional trends in heatwave characteristics.** Our employment of Berkeley Earth allows for the first comprehensive assessment of regional changes in observed heatwaves (Table 1, Supplementary Table 2), where adequate data exist for almost all of the 26 SREX regions (see "Methods", exception is Canada, Greenland, Iceland; CGI). During 1950−2017 (Table 1), the largest trends were mainly over low-latitude regions. At least one extra heatwave day has occurred each decade over the majority of regions between 1950 and 2017 (though this can be as high as 3−5 days decade$^{-1}$ over low-latitude areas), and heatwaves have increased in length by between 0.2 to over 1 day decade$^{-1}$. In general, regional trends beginning in a later decade can be larger in magnitude (Supplementary Table 2), which is at least in part due to greater overall warming of the global climate in later decades[22]. However later regional trends also bare less significance (Supplementary Table 2, Figs. 3c, d, 4d), due to the influence of internal climate variability at shorter timescales[20,23]. Only one region (Central North America; CNA) shows no significant change for all heatwave characteristics. These trends are in line with other reported temperature changes over this region[24,25].

While regional trends in heatwave frequency, maximum duration, and cumulative heat are mostly significantly increasing

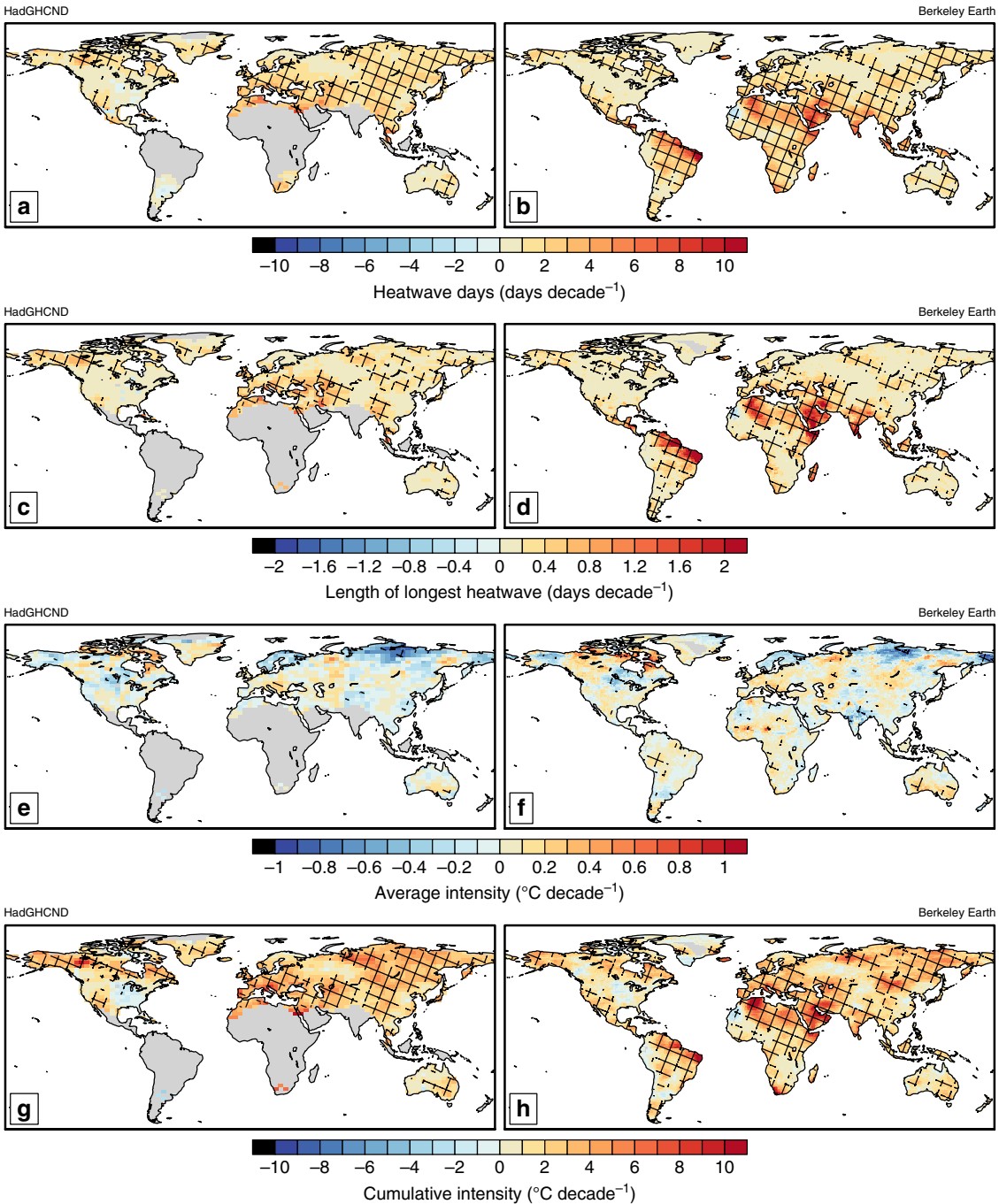

**Fig. 1 Global maps of observed decadal heatwave trends.** Trends in seasonal heatwave days (**a**, **b**); length of longest heatwave (**c**, **d**); average heatwave intensity (**e**, **f**); and cumulative heat (**g**, **h**) for quasi-global observational dataset HadGHCND (**a**, **c**, **e**, **g**) and global observational dataset Berkeley Earth (**b**, **d**, **f**, **h**) over the period 1950−2014. Trends are expressed as days decade$^{-1}$ for (**a**−**d**), and °C decade$^{-1}$ for (**e**−**h**).

(exception is CNA, see above), trends in average intensity are only significant for regions Amazon (AMZ), Mediterranean (MED), North East Brazil (NEB), Southeast South America (SSA) and West Asia (WAS). Nine regions, as well as the global average, have slightly decreasing (though non-significant) trends in average intensity. Therefore, it is unlikely that the regional changes in average intensity are responsible for larger, more significant increases in cumulative heat.

Although mostly insignificant, trends in average intensity are smaller (or as mentioned above, even decreased) for most regions than global warming over the same time period (0.1 °C decade$^{-1}$ since 1950)[26]. By itself, this result is not entirely surprising since

the heatwave definition used here is based on a fixed threshold (see "Methods"); thus, average intensity is inversely proportional to the number of heatwave days. Therefore, as the number of heatwave days increase (Fig. 1b, Table 1), little or no change in average intensity can be expected. However, measuring heatwave intensity in this manner does not address the fact that more heatwaves mean more overall exposure to extreme temperatures. Indeed, other well-used measures of heatwave intensity such as the hottest heatwave day[1,13,27]—which generally have more significant trends than average intensity—also do not account for this. Our assessment of cumulative heat fills this gap. Consistent with global changes (Fig. 1h), regional changes in cumulative heat have

**Table 1 Decadal trends in seasonal heatwave characteristics.**

| Region | Heatwave days | Maximum duration | Average intensity | Cumulative heat |
|--------|--------------|------------------|-------------------|-----------------|
| ALA | **1.76** | **0.38** | 0.11 | **4.11** |
| AMZ | **5.40** | **1.16** | **0.05** | **3.03** |
| CNA | **3.71** | **0.66** | −0.02 | **2.12** |
| CAS | **2.08** | **0.38** | 0.10 | **3.59** |
| CEU | **1.84** | **0.36** | 0.05 | **4.40** |
| CGI | **0.90** | **0.23** | 0.07 | **2.34** |
| CNA | 0.62 | 0.16 | −0.04 | 0.90 |
| EAF | **3.75** | **0.96** | 0.08 | **3.09** |
| EAS | **1.60** | **0.28** | 0.03 | **2.46** |
| ENA | **1.23** | **0.28** | 0.01 | **1.17** |
| MED | **2.61** | **0.61** | **0.13** | **4.24** |
| NAS | **1.27** | **0.24** | −0.04 | **3.56** |
| NAU | **1.37** | **0.32** | 0.00 | **1.66** |
| NEB | **5.98** | **1.49** | **0.06** | **4.32** |
| NEU | **1.23** | **0.23** | −0.13 | **2.92** |
| SAF | **3.04** | **0.60** | 0.03 | **3.65** |
| SAH | **3.16** | **0.61** | −0.01 | **3.56** |
| SAS | **4.78** | **1.21** | 0.05 | **3.43** |
| SAU | **1.02** | **0.16** | 0.02 | **1.87** |
| SSA | **1.27** | **0.29** | **−0.19** | **1.32** |
| SEA | **4.20** | **0.80** | 0.01 | **2.24** |
| TIB | **1.34** | **0.29** | −0.03 | **2.37** |
| WAF | **1.71** | **0.31** | −0.01 | **1.42** |
| WAS | **5.24** | **1.19** | **0.16** | **5.50** |
| WSA | **1.42** | **0.33** | −0.05 | **1.18** |
| WNA | **1.17** | **0.20** | 0.04 | **2.21** |
| World | **2.26** | **0.46** | −0.03 | **2.84** |

Heatwave days, maximum heatwave duration, average heatwave intensity and cumulative heat are presented for all 26 regions outlined in Supplementary Table 1, and the global average. Trends span 1950−2017, and appear in bold if significant at the 5% level.

increased since 1950. Trends range from ~1 °C to 4.5 °C decade$^{-1}$ and are significant for all but one region (CNA, discussed above). This demonstrates that the integration of anomalies over heatwave days (see "Methods") has resulted in a substantial increase in the overall extreme heat experienced during heatwaves across almost every region. This is a concern in regard to the many adverse heatwave impacts, where greater overall exposure bears more influence than changes in the intensity of a specific heatwave day, or the overall average[2,5,7–9,19,28,29].

**Cumulative heat analysis.** The highest amount of cumulative heat across all heatwaves for a given season varies regionally (Fig. 2a). For example, around an extra 80 °C of heat was experienced during the worst heatwave season over Australia, whereas over 240 °C of extra heat was felt during western Russia's worst season (Fig. 2a). The Mediterranean and Siberia have both experienced seasons where heatwaves contributed an extra 200 °C, and Alaska experienced its worst season where 150 °C extra heat was contributed (Fig. 2a). Perhaps unsurprisingly, much of the world's worst heatwave seasons as measured by cumulative heat have occurred since 2000, with the vast majority occurring since the 1980s (Fig. 2b).

Cumulative heat is the product of all seasonal heatwave days (i.e. heatwave frequency) and average heatwave intensity (see "Methods"). Thus, it is worth noting that the pattern and magnitude of trends in cumulative heat (Fig. 2c) looks very similar to that of heatwave frequency (Fig. 2d), when both characteristics are presented as percentage change (see "Methods"). Significant trends are biggest over northern South America, the Middle East, the Maritime Continent and much of Africa, at

50% decade$^{-1}$. However, significant trends between 10 and 30% decade$^{-1}$ exist over most other regions.

The average anomaly a given heatwave day contributes to cumulative heat shows a noteworthy latitudinal gradient (Fig. 2e). Heatwave days in high latitudes contribute, on average, 2−3 °C of extra heat day$^{-1}$ during a heatwave. In the Tropics, each heatwave day contributes between 0.5 and 1 °C. Moreover, during 1950−2017, trends in this value are quite small with patchy significance (Fig. 2f). However, regions that display the largest trends in cumulative heat (Figs. 1h and 2c) also have a significant trend in the average contribution of a heatwave day towards cumulative heat.

**How trends themselves are changing.** The heatwave trends discussed thus far are long-term changes over a period with substantial inter-regional and interannual variability, and intensifying global warming (Figs. 3a, b, 4a, b); so constant trends should not be assumed. Moreover, the combination of intensifying anthropogenic influence on the global climate, and the measurable influence of internal variability on heatwaves[20,23], strongly suggests that the trends themselves are not stable. However, previous assessments on the influence of internal variability on heatwaves has been limited to physical climate models[20,23]. For selected regions (see Supplementary Table 1), we present trends in each heatwave metric commencing in the years 1950−2000 inclusive and truncating in 2017 (Figs. 3c, d, 4c, d).

Trends in heatwave frequency, duration and cumulative heat (Figs. 3c, d, 4d) continue to increase in magnitude and remain significant when commencing in 1950 until at least 1970s. The exception is again CNA, where trends commencing in the early 1950s are not significant yet gain significance when commencing between the mid 1950s−1970s. This means that not only has the overall number of heatwave days, the length of the longest event and the extra heat experienced during heatwaves increased, but the speed at which these changes have occurred has accelerated during this timeframe. Note that for regions MED, WNA, East Asia (EAS), North Asia (NAS), and Northern Europe (NEU), as well as the global average, frequency and duration trends continue to accelerate up to trends commencing in the 1990s (Figs. 3c, d, 4d). Over MED, for example, heatwave frequency increases by over 2 days decade$^{-1}$ for trends commencing in 1950, accelerating to 6.4 days decade$^{-1}$ for trends commencing in the early 1980s (Fig. 3c). Acceleration of observed heatwave trends is not all that surprising given increasing global warming, and assessments of heatwave trends and individual events from climate models[27,30,31].

By the 1970s−1990s trends become non-significant and highly variable, which comes into play in some regions earlier than others. This further reflects the dominating influence of internal climate variability over heatwaves on multi-decadal timescales[20,23], even over more recent periods where the most amount of global warming has occurred[22]. We therefore advocate that any assessment of regional changes in heatwaves—irrespective of which heatwave metric is used and/or the amount of overall warming experienced—is measured over a period of at least 3−4 decades.

It is worth noting that with the exception of MED, most regionally and globally averaged average heatwave intensity trends (Fig. 4c) are not significant from 1950 onwards. This is consistent with the lack of trends in average heatwave intensity discussed above. Moreover, exceptional changes in extreme temperatures over MED have been outlined in previous studies[3,27,30,31]. Further, globally averaged trends in heatwave frequency, duration and cumulative heat are always significant until trends commence in 2000 or slightly later (Figs. 3c, d, 4d),

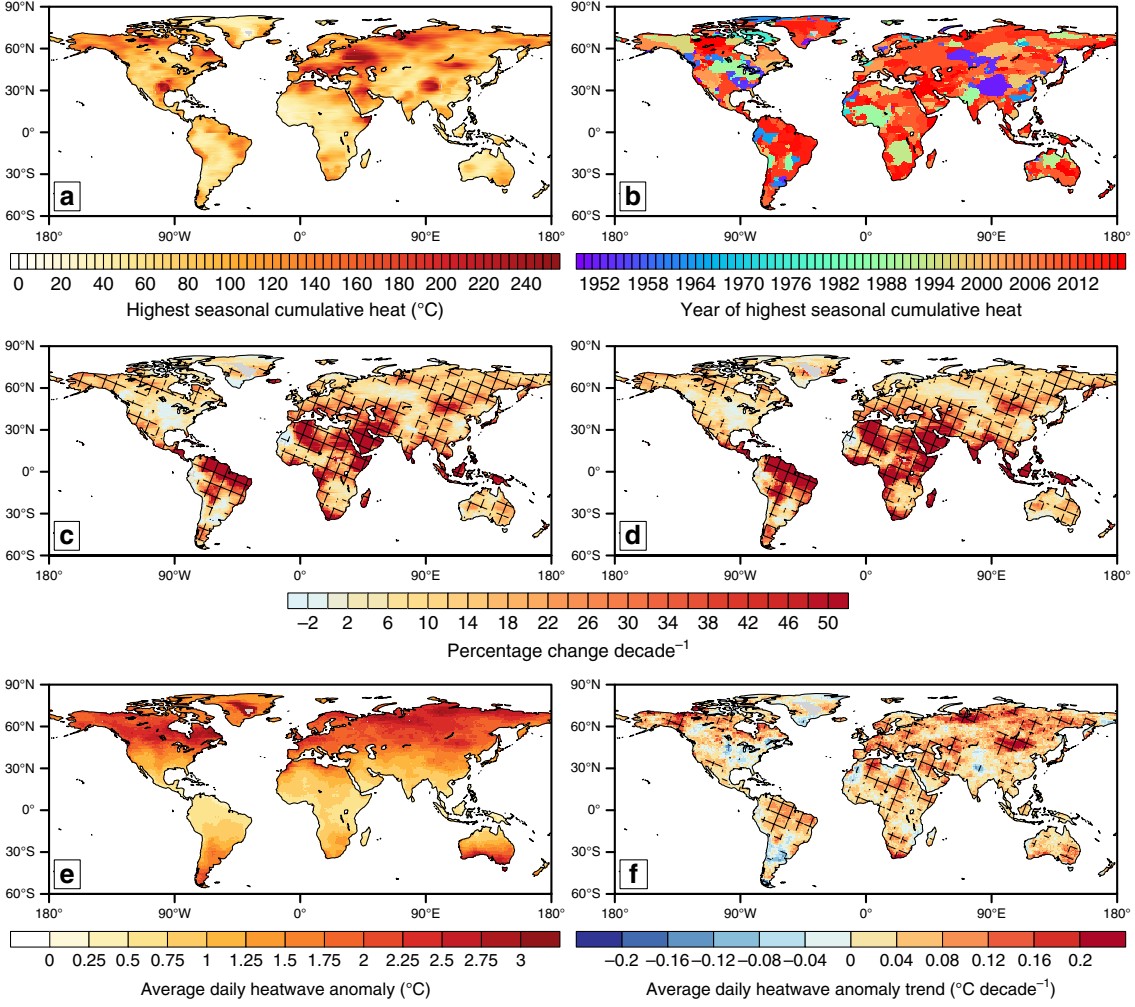

**Fig. 2 Global maps of cumulative heat statistics.** The highest seasonal cumulative heat (sum of anomalies relative to the calendar-day 90th percentile) (**a**); the year in which this value occurs (**b**); decadal trends in the percentage change of cumulative heat (**c**) and heatwave days (**d**); the average anomaly of a heatwave day (**e**) and the respective decadal trend (**f**). All values are calculated for the global observational dataset Berkeley Earth, for 1950−2017.

since the influence of variability is averaged out over larger spatial scales.

## Discussion

By employing a consistent heatwave measurement framework, our study presents the first comprehensive global and regional analysis of observed heatwave changes. Our findings demonstrate the concerning rate of heatwave increases since the mid-twentieth century. Moreover, trend magnitudes are not globally uniform, and are highest over regions known to experience disproportionately more adverse impacts of climate change[5,9]. While mainly significantly increasing trends since 1950 are evident for heatwave frequency, duration and cumulative heat, they are not monotonic. This study highlights that changes in heatwaves are not only increasing but accelerating in the presence of anthropogenic climate change. However, heatwaves are quite sensitive to internal climate variability[20,23], and regional trends shorter than a few decades are generally not reliable, even if the shorter period assessed includes the most recent acceleration of overall global warming[22]. Our study therefore recommends a period of at least 3−4 decades to robustly assess changes in heatwaves, which is considerably longer than that proposed for average temperature trends[21,32].

The cumulative heat—or the extra heat experienced during heatwaves—has markedly increased both globally (Fig. 1h) and

regionally (Table 1, Fig. 4b, d). We demonstrate this is largely driven by increases in the overall number of heatwave days, although over some regions slight increases in average intensity also contribute. Attempts to provide a measure of accumulated heat have been suggested previously[19], yet ours is the first to do so at both regional and global scales for the historical record. Moreover, our measure is likely more suitable for a range of impacts, instead of just one sector[19]. Common metrics of heatwave intensity[1,13,27,32,33]—especially those in the climate community—either focus on a single day and therefore do not represent overall intensity changes or are hindered by averaging over a larger sample size. We therefore advocate that measures of heatwave intensity assess the extra heat generated by heatwaves, such as the metric proposed in this study.

The results of our study have important implications for all systems affected by chronic heat exposure. This is because the appropriate management and adaptation of these systems is influenced by the separate components that drive the overall change. For example, longer, slightly warmer heatwaves may require different management strategies across various sectors such as public health[2,10] and energy supply[8] than shorter and more intense events despite a similar change in cumulative intensity. However, more research is required on this topic. Future research should also investigate how changes in the decomposition of cumulative heat—as well as overall changes—

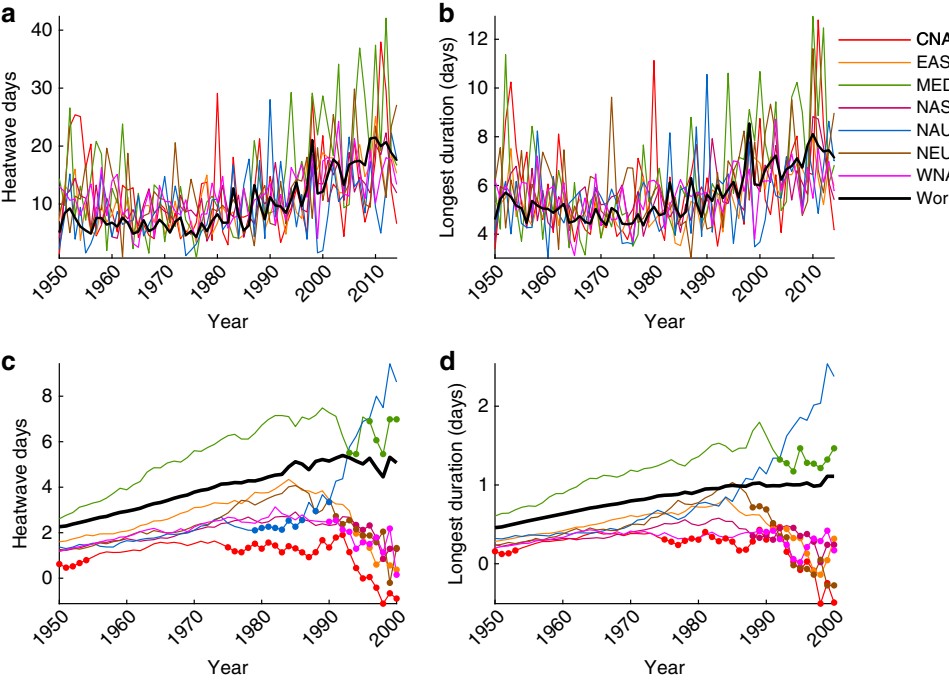

**Fig. 3 Heatwave regional and global timeseries.** Selected timeseries of heatwave days (**a**); and length of longest heatwave (**b**). Decadal trends in heatwave days (**c**); and length of longest heatwave (**d**) commencing yearly between 1950 and 2000 and truncating in 2017. All values are calculated for the global observational dataset Berkeley Earth (see "Methods"). Closed circles in (**c**) and (**d**) indicate when trends commencing in that year are NOT statistically significant at the 5% level (see "Methods"). Regions displayed are Central North America (CNA), East Asia (EAS), Mediterranean (MED), North Asia (NAS), North Australia (NAU), North Europe (NEU), West North America (WNA), and the global average (Wor).

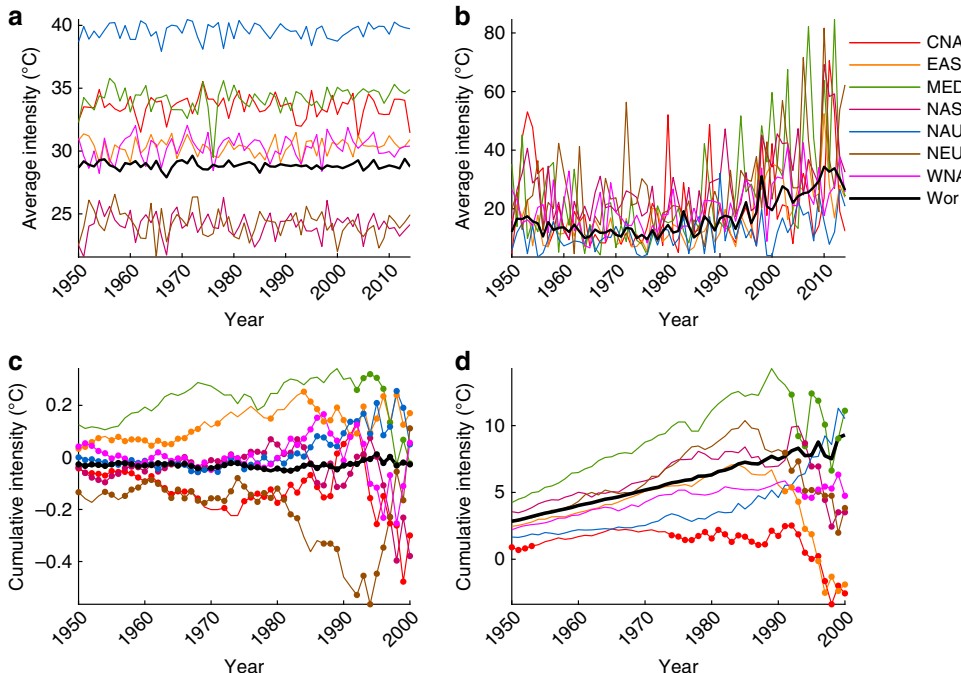

**Fig. 4 Heatwave regional and global timeseries.** Selected timeseries of average heatwave intensity (**a**); and cumulative heat (**b**). Decadal trends in average heatwave intensity (**c**); and cumulative heat (**d**) commencing yearly between 1950 and 2000 and truncating in 2017. All values are calculated for the global observational dataset Berkeley Earth (see "Methods"). Closed circles in (**c**) and (**d**) indicate when trends commencing in that year are NOT statistically significant at the 5% level (see "Methods"). Regions displayed are Central North America (CNA), East Asia (EAS), Mediterranean (MED), North Asia (NAS), North Australia (NAU), North Europe (NEU), West North America (WNA), and the global average (Wor).

filters down to impacts, as well as scrutinizing further acceleration in heatwave trends under increased anthropogenic climate change.

## Methods

**Data**. The bulk of our analysis is based on daily maximum temperatures ($T_{max}$) from the Berkeley Earth[18] gridded global land surface temperature dataset. Berkeley Earth is a relatively new global observational product on a 1° × 1° latitude/longitude grid, however, since at least the 1950s has been shown to perform within the bounds of other observed temperature products. The current version dates back to 1880 and uses a new mathematical framework such that the use of short and discontinuous datasets can be maximized. Input station data are weighted based on the quality and consistency they give to the spatial network. This algorithm allows Berkeley Earth to utilize five times more stations than other gridded observed temperature products. The lack of spatiotemporal gaps, especially since 1950, as well as its comparably high resolution makes Berkeley Earth an attractive dataset for assessing observed changes in temperature extremes such as heatwaves. This study is the first to demonstrate the capability of Berkeley Earth in the context of heatwave trends.

We also employ daily maximum temperatures from the HadGHCND dataset[12]. HadGHCND is also a gridded temperature dataset based on station data, interpolated to a 3.75° × 2.5° longitude/latitude grid. Because of the different criteria underpinning HadGHCND, less stations are used, which results in a quasi-global representation of observed temperature. However, because of its stricter quality control and homogenization of input stations, as well as its use in previous studies of observed changes in heatwaves, we use HadGHCND as a reference to determine the usefulness of Berkeley Earth in assessing global changes in heatwaves. HadGHCND currently extends from 1950 to 2014, and thus the two datasets are compared over this period.

**Heatwaves**. We employ the $T_{max}$ heatwave definition (CTX90)[1]. A heatwave is detected when at least 3 consecutive days are above the 90th percentile of $T_{max}$ for each calendar day. This percentile is based on a 15-day moving window of daily maximum temperatures over 1961−1990. Initially we look at three heatwave characteristics, intensity, frequency and duration. Heatwave frequency is defined as the sum of all heatwave days, heatwave intensity as the average intensity across all heatwave days, and duration as the longest event. Each characteristic is calculated for an extended summer season (May−September in the Northern Hemisphere, November−March in the Southern Hemisphere). All heatwaves are originally calculated at the grid box level. For regional analysis, seasonal metrics were spatially averaged according to the regional boundaries in Supplementary Table 1. We also introduce a fourth metric, cumulative heat. This describes the extra heat produced by heatwaves over a given season, and is the sum of the anomaly between each heatwave day and the calendar-day 90th percentile (i.e. the heatwave threshold) across all heatwave days in that season:

$$\text{heat}_{cum} = \sum_{1}^{n} T_{anom},\qquad(1)$$

where $\text{heat}_{cum}$ is the cumulative heat exposure, $n$ the number of heatwave days in a season, and $T_{anom}$ is the temperature anomaly relative to the calendar-day 90th percentile on a given heatwave day. This differs from previous assessments of excess heat in climate science that sum absolute temperature of days that exceed moderate extreme thresholds[34]. However, our metric only focuses on heatwave days[1], and the use of an anomaly focuses on the excess heat experienced once the heatwave threshold is exceeded. Using absolute temperatures during a heatwave can result in inflating heat exposure to a large measure that is not meaningful, as it is the exceedance of a threshold (over a number of consecutive days that constitutes a heatwave)—and not the fact that temperature is experienced in general—that results in excess heat exposure and adverse impacts. Therefore, we examine the anomaly of heatwave days relative to the heatwave threshold.

At the global scale, we also calculate the average anomaly of each heatwave day per season to determine whether the contribution of each day to $\text{heat}_{cum}$ is changing over time:

$$\text{avg}_{anom} = \frac{\text{heat}_{cum}}{\text{HWF}},\qquad(2)$$

where $\text{avg}_{anom}$ is the average temperature anomaly across all heatwave days in a given season, and HWF is the number of heatwave days in that season[8].

Also at the global scale, we assess changes in $\text{heat}_{cum}$ and HWF expressed as a percentage, to aid in a more direct comparison across the two metrics:

$$\text{change}_{\%} = 100\left[\frac{\text{seas}_{metric} - \text{metric}_{1961-1990}}{\text{metric}_{1961-1990}}\right],\qquad(3)$$

where $\text{change}_{\%}$ is the seasonal percentage change, $\text{seas}_{metric}$ is the respective heatwave metric for a given season, and $\text{metric}_{1961-1990}$ is the average of the respective metric over the base period (note this is the same base period as the calendar-day 90th percentile, from which heatwaves are calculated).

**Trend calculations**. Decadal trends were calculated via Sen's Kendal slope estimator[35], which is nonparametric and robust against outliers. Statistical significance was computed at the 5% level. Trends were calculated when at least 70% of daily temperature data were present per grid box, 9 years of which occurred after 2000. This condition is especially important for HadGHCND, which has inconsistencies in space and time for the period analysed, and where underpinning stations for some regions are no longer used after the new millennium. For comparison across datasets, trends are calculated between 1950 and 2014. The rest of analysis which only considers Berkeley Earth has trends calculated for 1950−2017. At the regional level, trends were calculated commencing each year between 1950 and 2000. Since only Berkeley Earth was used for regional analysis, all regional trends truncate in 2017.

## Data availability

The data underpinning this study are publicly available under the following DOIs: https://doi.org/10.6084/m9.figshare.12017847; https://doi.org/10.6084/m9.figshare.12017838; https://doi.org/10.6084/m9.figshare.12017811. The HadGHCND dataset is freely available at https://www.metoffice.gov.uk/hadobs/hadghcnd/. The Berkeley Earth dataset is freely available at http://berkeleyearth.org/.

## Code availability

The code used to calculate heatwaves is freely available at: https://doi.org/10.6084/m9.figshare.12021171.

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

## Acknowledgements

S.E.P.-K. is supported by ARC grant number FT170100106.

## Author contributions

S.E.P.-K. conceived the study, analysed the data and contributed to writing and manuscript editing. S.C.L. contributed to designing the study and with manuscript writing and editing.

## Competing interests

The authors declare no competing interests.
