## [Peer Review File · Nature Communications]

Reviewers' comments:

Reviewer #1 (Remarks to the Author):

Comments on the manuscript entitled "A comprehensive assessment of observed heatwave trends" by Perkins-Kirkpatrick and Lewis.

Comments

The authors present a study that examines changes in observed heatwaves across global regions. To this end, they compute trends across different time periods since 1950 so as to establish whether heatwaves are changing. In particular heatwave frequencies are exhibiting an overall increase. The authors also examine changes in cumulative exposure of heatwaves and the factors that are driven these changes. In my opinion, this is a nice study. Problematics that the work addresses are well explained. The aims of the study are well-suited for a journal as Nature Communications. The authors use appropriate and robust databases and methods to derive their results. The scientific goals fit well within the scope of this Journal. So, I would recommend publication. However, I feel that some scientific concerns should be first addressed prior to publication.

Specific remarks

1. Results. Figure 1 is repeated twice. The authors decompose cumulative exposure as the product of seasonal heatwave frequency and average heatwave intensity. The latter meaning average heatwave temperature across the season. Increases in cumulative exposure are mainly a result of an increase in heatwave frequency, while intensity is reducing over time. The authors should develop the physical meaning of this finding and not just mentioned in the text.
2. Conclusions. The authors state that cumulative exposure has not been previously presented in the context of atmospheric heatwaves. However, they also state in the introduction that this metric provides additional and essential information for impacts as stated by reference 19. So, I would suggest to the authors better contextualize such statement when at least a prior study derived a similar metric for atmospheric heatwaves. In general, statements throughout the manuscript have to be careful and accurate, taking into account the recent research carried out in climate change impacts.
3. Methods. The authors introduce cumulative heat exposure (CME) as the sum of the intensity across all heatwave days in a given season. To my understanding, this is the sum of the daily maximum temperature for each heatwave day. I mean, the sum of all the temperatures for all the days under heatwave conditions. Am I right? Or do you refer to intensity as the hottest day of the hottest heatwave? In this case, CME would be overestimated and some additional remark should be included in the text. Could you please clarify this issue in the manuscript?
4. In addition, the authors decompose CME as the product of the total number of heatwave days and the average heatwave intensity over the season, that it is a different definition. This is a bit messy for the readers and we would appreciate further clarification in this metric. When calculating the percentage contribution of days under heatwave conditions and the average HW intensity, the authors compute these metrics in the denominator as a sum of two quantities of different nature and units. I understand that this is just the computation of the proportion of the two different variables contributing to CME, but I think that it also needs further clarification.

Reviewer #2 (Remarks to the Author):

Review for "A comprehensive assessment of observed heatwave trends" submitted to Nature Communications.

(by S.E. Perkins-Kirkpatrick and S.C. Lewis)

The authors analyze historical trend in heatwave duration, intensity and frequency by using different heatwaves indices calculated from two different global datasets. The manuscript is well written and report a clear message. However, many other studies have shown heatwave trend in the past, present and future climate. The findings presented in this study do not represent a novelty in the field. Even the heatwave exposure, as calculated here, has been defined and discussed in other studies. I would suggest to submit this study in other topical journals.

Specific comments:

- 1.26 Is it True? Many studies have analysed changes in observed heatwaves across global regions (See for example Frich et al 2002 and Alexander et al 2006).
- 1.27 the climate change signal is not linear with an intuitive consequence that the rate of heatwave change is not expected to be constant (associated to linear trend). Many studies quantify linear trend because it is the most simple way to estimate it. Moreover the linear trend found here is because the authors are calculating trend in a 60 years period. This represent a short time period for climate signal that is expected to change in a decadal scale. Just to give an example, it is like we estimate annual trend in a time series of six years.
- 1.275-276 summer in the tropics occur in different periods than the ones chosen in this study (May-September in the North and November-March in the South). Could we have some bias for this? please explain!
- 1.278 Is it the sum of temperature values across all heatwave days? If Yes please note that other studies have introduced this metric (see for example Russo and Sterl 2011)
- 1.283 equation 1: the variable defined here seems to be the same of the one above on heatwave exposure. According to what I have understood from this formula the authors multiply and divide for the same variable: $days_{HW}$. Is it? please specify.

Minor Comments

l.19 According to Nature Communications template text should not be referred in the abstract.

References

1. Frich, P., L. V. Alexander, P. DellaMarta, B. Gleason, M. Haylock, A. M. G. Klein Tank, and T. Peterson (2002), Observed coherent changes in climatic extremes during the second half of the twentieth century, *Clim. Res.*, 19(3), 193212.
2. Alexander, L. V., et al. (2006), Global observed changes in daily climate extremes of temperature and precipitation, *J. Geophys. Res.*, 111, D05109, doi:10.1029/2005JD006290.
3. Russo, S., and A. Sterl (2011), Global changes in indices describing moderate temperature extremes from the daily output of a climate model, *J. Geophys. Res.*, 116, D03104,

Reviewer #3 (Remarks to the Author):

This paper analyses trends in heatwave statistics (measures of heatwave length, intensity and frequency, and cumulative exposure) averaged for the globe and also several large domains, and is based on observational data.

Some, but not all, definitions of measures used are clear. My review is based on my understanding as below:

- A heatwave period occurs when at least 3 consecutive days exceed the 90th percentile of Tmax per calendar day, which is defined using a sliding window of 15 days centred on each calendar day, over the period 1961-1990.
- Frequency is the number of days satisfying the heatwave criterion per May-Sep (or Nov-March) summer season.
- Intensity is the hottest day of the hottest heatwave per summer season, measured in degrees Celsius (relative to 0 degC). Note this is not necessarily the hottest day of the year.
- Duration is the longest heatwave event per summer season.
- Cumulative heat exposure is the sum of intensity over all days satisfying the heatwave criterion per summer season. According to equation (1), average heatwave intensity is the same as cumulative heat exposure divided by the frequency of heatwave days per summer season. Note that results are only shown for relative average heatwave intensity, not the full value.

Below I list the main claims made by the manuscript, then discuss their novelty and some concerns about their validity, along with some suggestions.

The main claims are:

- (1) This is the first study that examines changes in observed heatwaves systematically across global regions.
- (2) The study is the first to show that changes in heatwaves for the chosen statistical measures are not only increasing but are accelerating in the presence of anthropogenic climate change.
- (3) For the first time, they scrutinize (decompose) changes in cumulative exposure.
- (4) That the increase in cumulative exposure is mainly driven by the increasing contribution of heatwave days, and that the influence of average heatwave intensity on cumulative exposure has declined.
- (5) Results provide essential information for impacts, on the spatial and temporal scales that are necessary.
- (6) Post 1970s/1980s, internal climate variability is more dominant over heatwave changes, and thus heatwave changes over this period – as well as other periods of similar length – should not be used as reliable indicators of heatwave changes.

Novelty

The authors state that claims 1-3 above are novel.

- (1) The authors mention themselves another study that attempted to address heatwave changes globally, but say that the Berkeley data set they use here is more spatially complete and superior in

resolution at a scale useful for impacts. I am not aware of other studies with the same aim.

(2,3) I am not aware of any other studies that 'scrutinize' changes in cumulative exposure. However there are other studies that show non-linear increases in time of heatwave intensity and other parameters over regional or large domains for the same time period (from 1950's onwards) e.g. in Stott et al. (2004), JJA temperature anomalies in a Mediterranean domain. WWA studies e.g. Fig 2a of Kew et al. (2019), use a linear fit to model changes in observed heatwave intensity but using global mean surface temperature as a covariate, not time. That the increase is not at a 'constant rate in time' is not surprising. It would be interesting to see how figures 3 and 4 would change if plotted against global mean surface temperature instead of time.

Concerns on validity and suggestions

Concerning (1), whilst it is commendable to carry out a systematic analysis, the impact of exceeding a percentile-based definition of heatwaves varies across climate regions with different variability. For instance, in tropical regions with small temperature variability, a small exceedance of the 90th percentile can be a small anomaly in absolute temperature and might not have much of an impact.

Concerning (3 and 4), whilst the approach is novel, and the trend in cumulative exposure is interesting from an impacts point of view, it is not so convincing that the relative contribution of average heatwave intensity to cumulative exposure is relevant for impacts. Does the decreasing contribution of average intensity have any meaning for impacts? Isn't the increasing absolute intensity more relevant? It is also misleading that the authors occasionally refer to the contribution of average intensity to cumulative heat exposure as a contribution of intensity (e.g. L279).

The authors also need to give claim (4) further thought. The validity of this claim and significance is not (yet) convincing. My concern is that the result - that changes in cumulative exposure are mainly driven by changes in frequency - might be expected from the definitions used.

The intensity of heatwaves, defined as the hottest day of the hottest heatwave per year, is found to increase. However, a second measure of intensity, on which the claim (4) is based, is an average intensity, which is the cumulative exposure per season divided by heatwave frequency. As global warming continues, it is expected that the number of days with temperatures exceeding a fixed (per calendar day) threshold will increase. However, by construction, average intensity is inversely proportional to the number of heatwave days, so obviously this weighs negatively against the contribution from any increase in absolute intensity. They authors find that the influence of average heatwave intensity on cumulative exposure has declined.

In addition, the possible change in relative contribution of mean intensity and number of heatwave days depends on their definitions. Intensity is (arbitrarily) measured in degrees C above freezing point, so this can only increase by a few percent e.g. 30 to 32 degC is a change of about 7% in itself, which then must be divided by heatwave frequency. The heatwave frequency, however, can easily double or triple. The world curve shows a range from around 5 to 20 - an increase of 300%. Is this physically possible to match in (average) intensity with the chosen definition?

The authors could choose to simply present the cumulative exposure analysis without the decomposition and remove the claim - i.e. omit use of equations (2) and (3) in the analysis. Otherwise, they should prove that the result is not simply expected by construction, or how the results differ from a null-hypothesis. The authors could consider, for example, what happens to the relative contributions to cumulative heat exposure trends if there is no change in variability and a linear increase in temperature (intensity)? How large would e.g. the background trend (or increase in variability) need

to be for the 'average intensity' to contribute more to cumulative exposure than frequency? Is that realistically possible? They should also discuss how the sensitivity of the outcome to their heatwave definition. If the authors want to say something about how average intensity is changing, they should consider an independent measure, in which the denominator does not change. e.g. average intensity of hottest X heatwave days of the summer months, with X a constant.

Concerning claim (5): The claim that the results are relevant and give essential information for impacts seems to be based on (i) using a data set with resolution 1deg x 1deg, which is indeed an improvement on the HadGHCND data set used for comparison (3.75deg x 2.5deg), and (ii) including cumulative heat exposure as an index. Impacts can sometimes be very specific and local. Saying 'relevant' is appropriate, but 'essential' is perhaps too strong.

Concerning claim (6): To me, this statement says that internal climate variability is relatively more important towards the end of the time series than it is at the beginning, but this information cannot be easily extracted from the figures as the time period used to assess the trends decreases along the time axis. Decadal trends are plotted against their starting year, "commencing yearly between 1950-2000, and truncating in 2017". This means the trend plotted at 1950 is the rate of change per decade estimated over the years 1950-2017, whilst the trend plotted at 2000 is the rate of change per decade estimated over the years 2000-2017. If this is correct, then it is not surprising that internal climate variability becomes more important as the sampling period is shortened. Neither is it surprising that the curves flatten off, as the latter decades are past the "hook" of the hockey stick - past the point of most rapid acceleration of change. Alternatively, if decadal trends for a 20-year period at the beginning of the series were compared to a 20-year period at the end, I would expect an opposite kind of statement, i.e. the trend to be more significant over internal climate variability for the latter 20-year period. Again, it would be interesting to see how figures 3 and 4 would change if plotted against global mean surface temperature instead of time, as we know the global mean surface temperature time series also has the hockey-stick form. How much of the increasing dominance of internal climate variability is due to the shortening time series?

Overall, the manuscript could still be interesting to others in the field, for its global systematic approach, as long as caveats and sensitivities are discussed. The text could be improved in clarity in many places, but I would first recommend that the authors consider the validity and relevance to impacts of their decomposition of cumulative exposure, as well as the discussing the sensitivity of results on their choice of definitions, baselines and averaging intervals.

References

Stott, P. A., D. A. Stone, and M. R. Allen, 2004: Human contribution to the European heatwave of 2003. *Nature*, 432, 610–614.

Kew, S. F., S. Y. Philip, G. J. van Oldenborgh, F. E. L. Otto, R. Vautard, and G. van der Schrier, 2019: The exceptional summer heatwave in southern Europe 2017 [in "Explaining Extremes of 2017 from a Climate Perspective"]. *Bull. Amer. Meteor. Soc.*, 100 (1), S49–S53, <https://doi.org/10.1175/BAMS-D-18-0109.1>.

Reviewer #1 (Remarks to the Author):

Comments on the manuscript entitled “A comprehensive assessment of observed heatwave trends” by Perkins-Kirkpatrick and Lewis.

Comments

The authors present a study that examines changes in observed heatwaves across global regions. To this end, they compute trends across different time periods since 1950 so as to establish whether heatwaves are changing. In particular heatwave frequencies are exhibiting an overall increase. The authors also examine changes in cumulative heat of heatwaves and the factors that are driven these changes. In my opinion, this is a nice study. Problematics that the work addresses are well explained. The aims of the study are well-suited for a journal as Nature Communications. The authors use appropriate and robust databases and methods to derive their results. The scientific goals fit well within the scope of this Journal. So, I would recommend publication. However, I feel that some scientific concerns should be first addressed prior to publication.

We thank the reviewer for their kind words and encouragement

Specific remarks

1. Results. Figure 1 is repeated twice. The authors decompose cumulative heat as the product of seasonal heatwave frequency and average heatwave intensity. The latter meaning average heatwave temperature across the season. Increases in cumulative heat are mainly a result of an increase in heatwave frequency, while intensity is reducing over time. The authors should develop the physical meaning of this finding and not just mentioned in the text.

Apologies for the figure repetition – it should be fixed in this resubmission.

*We have updated the calculation of the cumulative heat, based on suggestions by this reviewer and #3. We have now calculated it as the sum of anomalies over all heatwave days in a season, with the anomaly relative to the calendar-day 90th percentile (see heatwave definition in Methods). This is a better representation of the **extra** heat (rather than the total heat) that a heatwave brings, separate to that of a non-heatwave day. Upon reflection of the initial metric we found that whether or not a day is defined as part of a heatwave can differ by a fraction of a degree. Thus summing absolute temperatures across all heatwave days could lead to very large measures of cumulative heat that are not all that meaningful. For example a heatwave day where the temperature reaches 34.5°C and the 90th percentile is 34°C now contributes 0.5°C to the cumulative heat, as opposed to the full 34.5°C, which is similar to the temperature (and therefore the total heat) of a non-heatwave day of 33.9°C that is left out of the cumulative heat calculation. Summing the anomalies relative to the 90th percentile still includes the necessary days without over-inflating the cumulative heat, while ensuring that the metric is more representative to the local climate. The new version of cumulative heat is much more meaningful and useful, as it is the **extra** heat a hot day brings that causes adverse impacts, not the total heat.*

As mentioned by reviewer #3 the average intensity of a heatwave is inversely proportional to the number of heatwave days – we have now explained and expanded on this in the text. IE, that the reason why we are seeing increases in cumulative heat where intensity remains mainly unchanged is because we are seeing more heatwave days. More days now exceed the heatwave threshold (calendar-day 90th percentile between 1961-1990), which therefore means more cumulative heat is generated.

2. Conclusions. The authors state that cumulative heat has not been previously presented in the context of atmospheric heatwaves. However, they also state in the introduction that this metric provides additional and essential information for impacts as stated by reference 19. So, I would suggest to the authors better contextualize such statement when at least a prior study derived a similar metric for atmospheric heatwaves. In general, statements throughout the manuscript have to be careful and accurate, taking into account the recent research carried out in climate change impacts.

We agree with the reviewer on this issue. We have better contextualised our study in terms of the background literature, and reworded our conclusion so that the novelty of our work is clear but at the same time not overstated.

3. Methods. The authors introduce cumulative heat exposure (CME) as the sum of the intensity across all heatwave days in a given season. To my understanding, this is the sum of the daily maximum temperature for each heatwave day. I mean, the sum of all the temperatures for all the days under heatwave conditions. Am I right? Or do you refer to intensity as the hottest day of the hottest heatwave? In this case, CME would be overestimated and some additional remark should be included in the text. Could you please clarify this issue in the manuscript?

The reviewer's former statement was correct. However as mentioned above we have reviewed and changed how we define CME (which we now just call "cumulative heat"), and reduce any overestimation.

In the previous manuscript version heatwave intensity (a separate metric) was defined as the hottest heatwave day. However, based on your comment below, and concerns from Reviewer #3 we now present heatwave intensity as the average intensity across all heatwave days in a season (see Methods; Perkins and Alexander, 2013). This makes heatwave intensity more comparable to CME and our new measure of cumulative heat.

4. In addition, the authors decompose CME as the product of the total number of heatwave days and the average heatwave intensity over the season, that it is a different definition. This is a bit messy for the readers and we would appreciate further clarification in this metric. When calculating the percentage contribution of days under heatwave conditions and the average HW intensity, the authors compute these metrics in the denominator as a sum of two quantities of different nature and units. I understand that this is just the computation of

the proportion of the two different variables contributing to CME, but I think that it also needs further clarification.

We agree with the reviewer that this was quite unclear, especially when combining quantities with different units. Instead of further clarification we have revised CME (see above) and based on comments from Reviewer #3 have removed the deconstruction of CME, at least in its previous form. Instead, in a revised version of Figure 2, we show how the average anomaly of a heatwave day (CME/heatwave days per season) displays much smaller, and mainly insignificant trends. Moreover, the percentage change in CME and heatwave days (we calculated percentage change for a more straightforward comparison) show clear similarities, thus supporting our argument that the increase of heatwave days is the main driver behind CME.

Reviewer #3 (Remarks to the Author):

This paper analyses trends in heatwave statistics (measures of heatwave length, intensity and frequency, and cumulative heat) averaged for the globe and also several large domains, and is based on observational data.

Some, but not all, definitions of measures used are clear. My review is based on my understanding as below:

- A heatwave period occurs when at least 3 consecutive days exceed the 90th percentile of Tmax per calendar day, which is defined using a sliding window of 15 days centred on each calendar day, over the period 1961-1990.
- Frequency is the number of days satisfying the heatwave criterion per May-Sep (or Nov-March) summer season.
- Intensity is the hottest day of the hottest heatwave per summer season, measured in degrees Celsius (relative to 0 degC). Note this is not necessarily the hottest day of the year.
- Duration is the longest heatwave event per summer season.
- Cumulative heat exposure is the sum of intensity over all days satisfying the heatwave criterion per summer season. According to equation (1), average heatwave intensity is the same as cumulative heat exposure divided by the frequency of heatwave days per summer season. Note that results are only shown for relative average heatwave intensity, not the full value.

Reviewer 3 makes correct assumptions here. However, as described in comments above to Reviewer #1, we have changed how cumulative heat exposure (now just "cumulative heat") is calculated. Moreover, to make comparison between cumulative heat and heatwave intensity more straightforward, we now use average heatwave intensity, instead of peak intensity.

Below I list the main claims made by the manuscript, then discuss their novelty and some concerns about their validity, along with some suggestions.

The main claims are:

(1) This is the first study that examines changes in observed heatwaves systematically across global regions.

(2) The study is the first to show that changes in heatwaves for the chosen statistical measures are not only increasing but are accelerating in the presence of anthropogenic climate change.

(3) For the first time, they scrutinize (decompose) changes in cumulative heat.

(4) That the increase in cumulative heat is mainly driven by the increasing contribution of heatwave days, and that the influence of average heatwave intensity on cumulative heat has declined.

(5) Results provide essential information for impacts, on the spatial and temporal scales that are necessary.

(6) Post 1970s/1980s, internal climate variability is more dominant over heatwave changes, and thus heatwave changes over this period – as well as other periods of similar length – should not be used as reliable indicators of heatwave changes.

The reviewer is correct in that these were the claims made by the original manuscript. In the revised manuscript, based on the suggestions of this reviewer and #1, we do no longer explore (3) or (4) in as much detail and moderate these claims. Moreover, we have re-worded how we addressed (6) as this was not precisely what was intended. Rather we meant to convey that internal variability effects heatwaves on shorter timescales, even over periods where more global warming has been experienced. Thus we suggest using periods of at least 3-4 decades to estimate changes in heatwaves, even if this is a period of larger overall warming such as from the 1970s/1980s onwards.

Novelty

The authors state that claims 1-3 above are novel.

(1) The authors mention themselves another study that attempted to address heatwave changes globally, but say that the Berkeley data set they use here is more spatially complete and superior in resolution at a scale useful for impacts. I am not aware of other studies with the same aim.

(2,3) I am not aware of any other studies that ‘scrutinize’ changes in cumulative heat. However there are other studies that show non-linear increases in time of heatwave intensity and other parameters over regional or large domains for the same time period (from 1950’s onwards) e.g. in Stott et al. (2004), JJA temperature anomalies in a Mediterranean domain. WWA studies e.g. Fig 2a of Kew et al. (2019), use a linear fit to model changes in observed heatwave intensity but using global mean surface temperature as a covariate, not time. That the increase is not at a ‘constant rate in time’ is not surprising. It would be interesting to see how figures 3 and 4 would change if plotted against global mean surface temperature instead of time.

We agree that the non-linear change of heatwaves over the time period analysed is not surprising. To make this clearer, we have now cited Stott et al 2004 and Kew et al 2019, and that our study is in agreement with other literature on this matter. We agree that plotting changes against GMST increase would be interesting, but it is outside the scope of this study (IE a more complete and up-to-date assessment of observed heatwave changes; the introduction of cumulative heat and how this compares to the other heatwave metrics on which it is based). We now also reference Perkins-Kirkpatrick and Gibson 2017; and Seneviratne et al 2016 that describe how heatwaves and other temperature extremes scale against GMST from physical climate model projections.

Concerns on validity and suggestions

Concerning (1), whilst it is commendable to carry out a systematic analysis, the impact of exceeding a percentile-based definition of heatwaves varies across climate regions with different variability. For instance, in tropical regions with small temperature variability, a small exceedance of the 90th percentile can be a small anomaly in absolute temperature and might not have much of an impact.

This could be true, however another possibility is that inhabitants/ecosystems of a local (tropical) climate is in tune to a very small temperature range, thus exceedances of this range (and lots of them) could have devastating impacts. The revised version of the cumulative now sums across anomalies relative to the heatwave threshold (90th percentile calendar-day percentile) so is no longer absolute temperature, though as we can see from Figure 2e, the average anomalies of heatwave days are less in the Tropics than at higher latitudes. Regardless of this, we believe that a systematic analysis is still important.

Concerning (3 and 4), whilst the approach is novel, and the trend in cumulative heat is interesting from an impacts point of view, it is not so convincing that the relative contribution of average heatwave intensity to cumulative heat is relevant for impacts. Does the decreasing contribution of average intensity have any meaning for impacts? Isn't the increasing absolute intensity more relevant? It is also misleading that the authors occasionally refer to the contribution of average intensity to cumulative heat exposure as a contribution of intensity (e.g. L279).

*The reviewer has a point that the relative contribution may not be so interesting for impacts. But this was not the point of deconstructing cumulative heat, rather it was to determine and explain what was mainly responsible for the mainly significantly increasing trends in cumulative heat. While we have now removed the original deconstruction of the cumulative heat (based on concerns from reviewer #1 and #3) we still display the change in the anomaly per heatwave day (IE cumulative heat /heatwave days per season; Fig 2e)). Furthermore, cumulative heat now assesses how much **extra** heat accumulates over heatwaves (see response above to reviewer #1). We believe this is a better approach than summing absolute temperature for 2 main reasons, 1) two days may differ very little in their absolute intensity yet one may be considered in a heatwave (and thus cumulative heat) while the other isn't, thus inflating the final cumulative heat value; 2) It is the exceedance of a threshold that normally induces impacts (ref), not the overall absolute value.*

Our removal of the contribution assessment now precludes our references to intensity/average intensity to cumulative heat.

The authors also need to give claim (4) further thought. The validity of this claim and significance is not (yet) convincing. My concern is that the result - that changes in cumulative heat are mainly driven by changes in frequency - might be expected from the definitions used.

The intensity of heatwaves, defined as the hottest day of the hottest heatwave per year, is found to increase. However, a second measure of intensity, on which the claim (4) is based, is an average intensity, which is the cumulative heat per season divided by heatwave frequency. As global warming continues, it is expected that the number of days with temperatures exceeding a fixed (per calendar day) threshold will increase. However, by construction, average intensity is inversely proportional to the number of heatwave days, so obviously this weighs negatively against the contribution from any increase in absolute intensity. They authors find that the influence of average heatwave intensity on cumulative heat has declined.

We agree with their reviewer on their points here, especially in how average intensity is inversely proportional to heatwave days. We have now included this as part of our discussion in the manuscript. However we do not think this affects the validity of claim 4. While changes in global warming are responsible for increases in heatwaves (see Perkins-Kirkpatrick and Gibson 2017), the extra heat exposure we are experiencing during heatwaves is mainly because there are more heatwave days, with little influence from changes in (average) heatwave intensity (note that, as described above, average heatwave intensity is now used instead of peak heatwave intensity such that comparisons with cumulative heat are more straightforward). We have made this clearer in the manuscript. Because – by construction – average intensity is inversely proportional to the number of heatwave days, little change or even a decrease can be expected. However it is important to make the point that we are still experiencing extra heat during heatwaves, which is what the cumulative heat quantifies, and this is largely driven by the increasing number of heatwave days.

In addition, the possible change in relative contribution of mean intensity and number of heatwave days depends on their definitions. Intensity is (arbitrarily) measured in degrees C above freezing point, so this can only increase by a few percent e.g. 30 to 32 degC is a change of about 7% in itself, which then must be divided by heatwave frequency. The heatwave frequency, however, can easily double or triple. The world curve shows a range from around 5 to 20 - an increase of 300%. Is this physically possible to match in (average) intensity with the chosen definition?

The reviewer makes an exceptional point here. We have now removed the relative contribution analysis (based on a comment made below). We have also included extra panels to Figure 2 (panels c and d, respectively) that show the percentage change in cumulative heat and heatwave days (see Methods) so that they are more comparable. Note the likeness in trends between the two measures, compared to a smaller, less significant change in the average anomaly of a heatwave day (CME/heatwave days per season; Figure 2f).

The authors could choose to simply present the cumulative heat analysis without the decomposition and remove the claim - i.e. omit use of equations (2) and (3) in the analysis. Otherwise, they should prove that the result is not simply expected by construction, or how the results differ from a null-hypothesis. The authors could consider, for example, what happens to the relative contributions to cumulative heat exposure trends if there is no change in variability and a linear increase in temperature (intensity)? How large would e.g. the background trend (or increase in variability) need to be for the 'average intensity' to contribute more to cumulative heat than frequency? Is that realistically possible? They should also discuss how the sensitivity of the outcome to their heatwave definition. If the authors want to say something about how average intensity is changing, they should consider an independent measure, in which the denominator does not change. e.g. average intensity of hottest X heatwave days of the summer months, with X a constant.

We feel we have adequately addressed this comment in previous comments to both reviewer #1 and #3. Note that the percentage changes in Figures 2c and 2d are relevant to the respective averages over 1961-1990 (IE a constant denominator).

Concerning claim (5): The claim that the results are relevant and give essential information for impacts seems to be based on (i) using a data set with resolution 1deg x 1deg, which is indeed an improvement on the HadGHCND data set used for comparison (3.75deg x 2.5deg), and (ii) including cumulative heat exposure as an index. Impacts can sometimes be very specific and local. Saying 'relevant' is appropriate, but 'essential' is perhaps too strong.

We have taken this onboard and reduced the strong emphasis. Thank you for pointing this out.

Concerning claim (6): To me, this statement says that internal climate variability is relatively more important towards the end of the time series than it is at the beginning, but this information cannot be easily extracted from the figures as the time period used to assess the trends decreases along the time axis. Decadal trends are plotted against their starting year, "commencing yearly between 1950-2000, and truncating in 2017". This means the trend plotted at 1950 is the rate of change per decade estimated over the years 1950-2017, whilst the trend plotted at 2000 is the rate of change per decade estimated over the years 2000-2017. If this is correct, then it is not surprising that internal climate variability becomes more important as the sampling period is shortened. Neither is it surprising that the curves flatten off, as the latter decades are past the "hook" of the hockey stick - past the point of most rapid acceleration of change. Alternatively, if decadal trends for a 20-year period at the beginning of the series were compared to a 20-year period at the end, I would expect an opposite kind of statement, i.e. the trend to be more significant over internal climate variability for the latter 20-year period. Again, it would be interesting to see how figures 3 and 4 would change if plotted against global mean surface temperature instead of time, as we know the global mean surface temperature time series also has the hockey-stick form. How much of the increasing dominance of internal climate variability is due to the shortening time series?

While increases relative to GMST is interesting, it is not in the scope of this study (please see response above). We do see the reviewer's point regarding 20-year trends at the end of the

period being less influenced by internal variability, however note that for most regions in Figures 3 and 4, few trends are significant beyond 1990. This suggests that, regardless of relative significance, internal variability is still having a measurable effect on changes over these relatively shorter periods, despite a more rapid rate of underlying warming occurring. Thus the message here is to exercise some caution when interpreting shorter-term heatwave trends (following from Perkins-Kirkpatrick et al 2017) as they are still quite vulnerable to internal variability, where fluctuations dampen the underlying signal. It is not all that surprising that shorter timeframes have a higher influence from variability, however our results here differ from previous assessments of average temperature trends, where significance is clear in timeframes of about 17 years. We have augmented the text to discuss these points.

Overall, the manuscript could still be interesting to others in the field, for its global systematic approach, as long as caveats and sensitivities are discussed. The text could be improved in clarity in many places, but I would first recommend that the authors consider the validity and relevance to impacts of their decomposition of cumulative heat, as well as the discussing the sensitivity of results on their choice of definitions, baselines and averaging intervals.

We appreciate the reviewer's constructive feedback and have sustainably edited the analysis based on their comments. We believe this has greatly improved the quality of the study, as reflected in the revised manuscript.

Reviewer #2

The authors analyze historical trend in heatwave duration, intensity and frequency by using different heatwaves indices calculated from two different global datasets. The manuscript is well written and report a clear message. However, many other studies have shown heatwave trend in the past, present and future climate. The findings presented in this study do not represent a novelty in the field. Even the heatwave exposure, as calculated here, has been defined and discussed in other studies. I would suggest to submit this study in other topical journals.

This is at odds with the reviews above. However we hope that our substantial edits have made the purpose of our study clearer.

- I.26 Is it True? Many studies have analysed changes in observed heat-waves across global regions (See for example Frich et al 2002 and Alexander et al 2006).

Yes – these studies, and also more modern ones (Perkins et al 2012) have been hampered by large gaps in observational datasets, so it has been difficult to truly assess global changes. This is why we compare our results to the HadGHCND dataset in Figure 1 – this is the dataset used by previous studies but has large gaps over many regions. Moreover the heatwave definitions use by Frich et al 2002 and Alexander et al 2006 have been shown to be non-representative of heatwaves in general (Perkins and Alexander, 2013).

- I.27 the climate change signal is not linear with an intuitive consequence that the rate of heatwave change is not expected to be constant (associated to linear trend). Many studies quantify linear trend because it is the most simple way to estimate it. Moreover the linear trend found here is because the authors are calculating trend in a 60 years period. This represent a short time period for climate signal that is expected to change in a decadal scale. Just to give an example, it is like we estimate annual trend in a time series of six years.

Linear trends can indeed be calculated over any time period. However, we do not just calculate trends over a 60+ year period, we also calculate trends over shorter time periods. We demonstrate that heatwave trends on short time periods inclusive of that where global warming has ramped up, are not significant, thus we caution the use of trends over shorter timescales for heatwaves. This is in line with previous assessments of how internal variability affects heatwaves (Perkins and Fischer, 2013; Perkins-Kirkpatrick et al 2017). Therefore we strongly discourage trends over periods as short as 6 years, and instead suggest they should be measured over a period of decades. We do not make assumptions that trends are constant since global warming itself is not constant, but this does not justify the use of short timescales for estimating trends.

- I.275-276 summer in the tropics occur in different periods than the ones chosen in this study (May-September in the North and November-March in the South). Could we have some bias for this? please explain!

Previous assessments of seasonal temperature extremes have used similar time periods to represent summer, with some even just considering a 3-month window. Moreover, it could be argued that Tropical regions do not experience “summer” at all. Although interesting and worthwhile, exploring whether more traditional seasonal timeframes produce biases on extreme temperatures and/or heatwaves is outside the scope of this study

- I.278 Is it the sum of temperature values across all heatwave days? If Yes please note that other studies have introduced this metric (see for example Russo and Sterl 2011)

Is the study you are referring to titled “Global change in indices describing moderate temperature extremes from the daily output of a climate model?” (doi:10.1029/2010JD014727). If so a cumulative measure of heat is definitely given, yet not for heatwaves, just all exceedances of the 90th percentile. Moreover, it as an assessment from one climate model, not observed trends. We have now cited this paper and based on the above reviewer comments have augmented how we measure cumulative heat (now an anomaly). We have also taken the emphasis off our study to produce measures of cumulative heat, but rather the first to do so for observed heatwaves at both regional and global scales.

- I.283 equation 1: the variable defined here seems to be the same of the one above on heatwave exposure. According to what I have understood from this formula the authors multiply and divide for the same variable: daysHW . Is it? please specify.

Please not we have now removed this part of the analysis, due to inconsistencies pointed out by the above reviewers.

REVIEWERS' COMMENTS:

Reviewer #1 (Remarks to the Author):

The authors have properly addressed the concerns pointed out by the reviewer in the revised version of the manuscript. Therefore, I would recommend publication.

Reviewer #3 (Remarks to the Author):

The authors have carefully considered all points raised and I am satisfied with their responses.

In my first review I mentioned that the text could be improved in clarity in many places and this is still the case. There are some sloppy mistakes and awkwardly written sentences. Please check the language thoroughly, or ask a native English speaker who is fresh to the paper to do so. I have listed below a few of the things I noticed.

Minor points

L28 abstr - some definition of cumulative heat should be included or make it clear that it will be defined later e.g. A measure of cumulative heat shows ...

L116 specify SREX - all of the 26 SREX regions

English in paragraph L128-133 needs improving.

L128 heat all -> heat are all

L130-131 not a sentence. presenting —> present

L131 It Is -> It is

L133 follows from?

L150 check english "likely concerning"? of concern? to cause concern?

Pay attention to spaces before/after commas and fullstops/periods.

Some phrases are overly complex e.g. L253-254. "likely has roots in the nature of what components drive"

L268-269. 'however since at least the 1950s has shown been to perform within the bounds of other observed temperature products.': shown been to perform —> been shown to perform

L316-318 "Thus it makes little sense to include the full or absolute temperature value of a given day, rather than the proportion of temperature that permits that day as part of a heatwave, which is the anomaly relative to the heatwave threshold." This is an uncomfortable sentence. I know what you want to say, but please improve the clarity and be scientifically accurate, e.g., how would you define a proportion of temperature? Is that a scientifically correct expression?

L343 important for

L344 during —> in

L347 have —> has